# Quantifying and Statistically Modeling Residual Pneumoperitoneum after Robotic-Assisted Laparoscopic Prostatectomy: A Prospective, Single-Center, Observational Study

**DOI:** 10.3390/diagnostics12040785

**Published:** 2022-03-23

**Authors:** Venkat M. Ramakrishnan, Tilo Niemann, Philipp Maletzki, Edward Guenther, Teodora Bujaroska, Olanrewaju Labulo, Zhufeng Li, Juliette Slieker, Rahel A. Kubik-Huch, Kurt Lehmann, Antonio Nocito, Lukas J. Hefermehl

**Affiliations:** 1Division of Urology, Brigham and Women’s Hospital, Harvard Medical School, Boston, MA 02115, USA; vramakrishnan@bwh.harvard.edu; 2Institute of Radiology, Kantonsspital Baden, 5404 Baden, Switzerland; tilo.niemann@ksb.ch (T.N.); rahel.kubik@ksb.ch (R.A.K.-H.); 3Division of Urology, Kantonsspital Baden, 5404 Baden, Switzerland; philipp.maletzki@ksb.ch (P.M.); kurt.lehmann@ksb.ch (K.L.); 4Department of Mathematics, Swiss Federal Institute of Technology, 8092 Zurich, Switzerland; edward.guenther@math.ethz.ch (E.G.); teodora.bujaroska@students.ethz.ch (T.B.); olabulo@student.ethz.ch (O.L.); zhufli@student.ethz.ch (Z.L.); 5Department of Surgery, Kantonsspital Baden, 5404 Baden, Switzerland; juliette.slieker@ksb.ch (J.S.); antonio.nocito@ksb.ch (A.N.)

**Keywords:** residual pneumoperitoneum, laparoscopy, prostatectomy, X-ray computed, robotic surgical procedures, carbon dioxide

## Abstract

Background: Laparoscopic surgery (LS) requires CO_2_ insufflation to establish the operative field. Patients with worsening pain post-operatively often undergo computed tomography (CT). CT is highly sensitive in detecting free air—the hallmark sign of a bowel injury. Yet, the clinical significance of free air is often confounded by residual CO_2_ and is not usually due to a visceral injury. The aim of this study was to attempt to quantify the residual pneumoperitoneum (RPP) after a robotic-assisted laparoscopic prostatectomy (RALP). Methods: We prospectively enrolled patients who underwent RALP between August 2018 and January 2020. CT scans were performed on postoperative days (POD) 3, 5, and 7. To investigate potential factors influencing the quantity of RPP, correlation plots were made against common variables. Results: In total, 31 patients with a mean age of 66 years (median 67, IQR 62–70.5) and mean BMI 26.59 (median 25.99, IQR: 24.06–29.24) underwent RALP during the study period. All patients had a relatively unremarkable post-operative course (30/31 with Clavien–Dindo class 0; 1/31 with class 2). After 3, 5, and 7 days, 3.2%, 6.4%, and 32.3% were completely without RPP, respectively. The mean RPP at 3 days was 37.6 mL (median 9.58 mL, max 247 mL, IQR 3.92–31.82 mL), whereas the mean RPP at 5 days was 19.85 mL (median 1.36 mL, max 220.77 mL, IQR 0.19–5.61 mL), and 7 days was 10.08 mL (median 0.09 mL, max 112.42 mL, IQR 0–1.5 mL). There was a significant correlation between RPP and obesity (*p* = 0.04665), in which higher BMIs resulted in lower initial insufflation volumes and lower RPP. Conclusions: This is the first study to systematically assess RPP after a standardized laparoscopic procedure using CT. Larger patients tend to have smaller residuals. Our data may help surgeons interpreting post-operative CTs in similar patient populations.

## 1. Introduction

In recent years, robotic-assisted laparoscopy has gained prominence across multiple surgical disciplines, such as general, vascular, thoracic, and urologic surgery [1,2,3,4]. As in conventional laparoscopy, the abdominal cavity is first filled with inert CO_2_ gas, which generates a working environment and establishes the surgical field. During the operation, the insufflated CO_2_ helps provide hemostasis but also results in increased intra-abdominal pressure, which increases airway pressures and systemic vascular resistance, decreases cardiac output and renal blood flow, and can result in hypercarbia [5]. At the end of surgery, the laparoscopic ports are extricated, and any insufflated gas is often manually compressed out of the abdomen prior to port closure. Despite this effort, it is inevitable that some fraction of insufflated gas will remain in the abdomen [6,7]. This is usually not an issue, as the insufflated gas is gradually absorbed and metabolized post-operatively.

Abdominal insufflation can, however, be an obfuscating factor when a post-operative patient presents to the hospital with abdominal pain. In such scenarios, computed tomography (CT) imaging is often employed as it is the gold standard for assessing peri- or post-operative complications and is very sensitive for detecting abdominal free air. This is a concerning finding potentially suggestive of a hollow-organ perforation [8]. While post-operative abdominal pain in this setting can be attributed to other etiologies such as visceral injury, vascular injury, infarction, peritonitis, or infection, the radiological and clinical interpretation of abdominal free air on a CT often poses a considerable problem [8]. It is often not possible to say with complete certainty whether the visualized air is actually residual insufflated gas from surgery (residual pneumoperitoneum, or RPP) or if it is attributable to more nefarious etiologies such as viscous perforation. 

Thus, this study aims to answer two relatively simple questions. First, how much free air (RPP) is expected in the post-operative setting after a robotic-assisted laparoscopic surgery? Second, does RPP correlate with demographic and operative data? Here, we attempt to systematically answer these questions in a popular urologic context—robotic-assisted laparoscopic prostatectomy (RALP)—using a series of CT scans obtained post-operatively. 

## 2. Materials and Methods

### 2.1. Study Design

We performed a prospective, single-center, observational study at our tertiary referral center. All patients undergoing RALP at our institution were eligible for inclusion in the study. Exclusion criteria consisted of (a) known visceral injury intra-operatively or (b) any visceral injury documented on the CT scan. Such criteria were employed as the overall aim of this study was not to evaluate the incidence or nature of post-operative complications in post-operative laparoscopic surgery patients, but rather to characterize RPP in patients with abdominal pain and an otherwise normal post-operative course. The study was approved by the local ethical commission. Informed and written consent was provided by all included patients. Given the descriptive intent of this study, we attempted to minimize the number of patients and number of post-operative CT scans (i.e., radiation exposure) per patient. Thus, the study size (31 patients) is in line with other similar studies (25 patients, plus an additional 20% to account for possible drop-outs) [6]. Patients were scanned on the 3rd, 5th, and 7th post-operative days (POD), with exams on POD 5 or POD 7 only taking place if any RPP was radiographically observed at the preceding time point(s).

All statistical analyses were performed using R: a language and environment for statistical computing (R Core Team, R Foundation for Statistical Computing, Vienna, Austria, 2020). An F-test (ANOVA) was performed for RPP comparison. For parameter estimation, a full Bayesian model was deployed [9]. For significance testing of the parameter estimates, a range of equivalence of plus or minus 0.1 standard deviations from the general population mean was used, as suggested by Kruschke and Liddell [10]. Unless otherwise specified, results were presented as means with corresponding medians and interquartile ranges (IQR). Results were considered significant if *p* < 0.05.

This research project was approved by the local ethics authorities (Project-ID: EKNZ 2018-00997) and conducted in accordance with the Declaration of Helsinki and the principles of Good Clinical Practice.

### 2.2. Surgery

RALP was performed using the daVinci Xi Surgical System (Intuitive Surgical, Sunnyvale, CA, USA) using a 12 mm AirSeal^®^ port and insufflator system (Conmed, Largo, FL, USA) with a standardized insufflation pressure of 12 mmHg. Changes in pressure were documented. After surgery, laparoscopic ports were removed, and manual compression of the abdomen was performed to subjectively release as much of the RPP as possible. All prostatectomies were radical in nature, and no drains were placed in any of the patients.

### 2.3. Parameters

We documented the patient’s age, size, weight, body mass index (BMI), operative duration, insufflation time, total insufflation volume, intra-abdominal pressure (average, maximum, minimum), manual abdominal decompression at the end of the case (yes/no), daily pain level and analgesic use, time of first flatulation, and any adverse events (within 7 days and within 30 days of surgery) via the Calvien–Dindo classification system [11]. 

### 2.4. CT Parameters, RPP Measurement, and Data Management

The high contrast difference between the intra-abdominal soft tissue, fat, and air allowed CTs to be performed via a non-contrast, low-dose protocol [12]. All images were analyzed by the leading CT radiologist at our institution. Of note: a total dose of approximately 3 mSv was anticipated for three CTs, which is below the average annual radiation exposure in Switzerland (approximately 4 mSv) [12]. Computer-aided 3D-volumetric segmentation of the free intra-abdominal gas was then performed using Syngo.via software (Siemens, Germany). The total RPP volume was then rendered in milliliters (mL). Figure 1 shows a sample image of thevolume measurement software interface (Figure 1).

## 3. Results

### 3.1. Demographic Data

In total, 31 men underwent a robotic-assisted laparoscopic prostatectomy (RALP) at our institution between August 2018 and January 2020. No significant blood loss requiring transfusion or major complications (Clavien–Dindo grades 4 and 5) occurred in any enrolled patient. A summary of the patient demographics and operative parameters is shown in Table 1.

The mean age of the patient population was 66.2 years (median 67, IQR 62–70.5), weight was 83.32 kg (median 83, IQR 75.5–90.5), and height was 177.16 cm (median 178, IQR 173–180.5), all leading to a BMI of 26.59 (median 25.99, IQR 24.06–29.24). The mean operative duration was 302.39 min (median 306, IQR 270–328) with a total insufflation time of 278.93 min (median 276.5, IQR 255–304.25), and total insufflation volume of 971.45 L (median 941, IQR 775–1242). 

Post-operatively, complications were relatively rare. In total, 30 of 31 patients received a Clavien–Dindo grade of 0 by POD 7, while one received a grade of 2 (for tachycardia, atrial fibrillation, and spontaneous conversion into sinus rhythm). First flatulence took place an average of 2.16 days post-operatively (median 2; IQR 2–3). 

Manual abdominal decompression was not performed on two patients. To this end, we expected that patients who did not receive manual decompression would demonstrate higher RPPs in the first follow-up post-operative CT scan compared to those who were decompressed. However, such a conclusion was not possible as the number of patients that were not decompressed was small (*n* = 2), and practically, the RPPs of non-decompressed patients on the first CT scan were not dramatically different from their decompressed counterparts.

### 3.2. Radiographic Assessment of Residual Pneumoperitoneum

The mean RPP volume (Table 1) in the first CT (POD 3) was 37.6 mL (median 9.58, max 247, IQR 3.92–31.82). By the second CT (POD 5), these figures decreased to 19.85 mL (median 1.36, max 220.77, IQR 0.19–5.61); and by the third (POD 7), RPP had nearly completely resolved, with a mean of 10.05 mL (median 0.09, max 112.42, IQR 0–1.5). 

At all three time points, (a) most patients (i.e., those at or below the 75th percentile) had RPP volumes less than approximately 30 mL, and (b) exhibited significant decreases in RPP (*p* < 4.8 × 10^−13^) between PODs 3 to 5 and 5 to 7 (Figure 2A). This trend was also seen across each individual patient (Figure 2B), though it is also easier to appreciate the high variance in volumes per person. Inversely, the number of patients without any detectable RPP expectedly rose exponentially over the same time frame (Figure 2C).

In total, the data demonstrate that while only a small handful of patients have relatively large-volume RPP, virtually all patients exhibit a near exponential decrease in the degree of RPP during the week following laparoscopic surgery.

### 3.3. RPP Correlation with Demographic and Operative Data

Next, RPP volumetric data were correlated with patient demographics (Figure 3). As seen in the figure, the correlation plot demonstrates the global trend of most variables being only weakly correlated. However, weight and BMI were negatively correlated with RPP volumes. 

Therefore, patient RPP volumes were plotted as a function of stratified BMIs, including those with normal BMIs, those who were overweight, or those who were obese (Figure 4). From this, obese patients always demonstrated significantly lower RPP volumes compared to their normally weighted and overweight counterparts (*p* < 2 × 10^8^). 

### 3.4. Generating and Understanding a Predictive Model for RPP

We calculated the RPP volume from the first, second, and third CT scans, which, again, corresponded with PODs 3, 5, and 7, respectively. The exponential decrease in RPPs over time (Figure 2A) in much of the patient cohort suggested a log-response model. As such, RPP volumes were log-transformed and put through a random-intercept-and-slope model to (a) capture the degree of RPP variation in our cohort and (b) generate a statistical model that could potentially be employed on future patients undergoing RALP [10]. As seen in Figure 2B, which depicts the RPP on a logarithmic scale per each individual patient, the RPP volumes of the first CT scan (i.e., the intercepts) and the rates of RPP decrease thereafter (i.e., the slopes) and are both highly variable. Our model allowed for correlated random effects, as it seemed that patients with higher RPPs on the first CT (i.e., a larger intercept) exhibited reduced rates of RPP resolution (i.e., shallower slopes) than patients with lower starting RPPs. Moreover, the model below incorporates the demographic findings from Figure 3 and Figure 4.
(1)$$y\sim N\left(\alpha_{j,\left[i\right]}+\beta_{j,\left[i\right]}x_{day,i}+\gamma x_{normalweight,i}+\delta x_{overweight,i}+\eta x_{obese,i^{\;}},\sigma_{y}{}^{2}\right),\;i\;=\;1,\ldots ,\;n;\;j\;=\;1,\ldots ,\;J$$
(2)(αj βj)~N((μα μβ),(σ2αρσασβρσασβσ2β))

Above, *y* is the log-transformed volume; *α_j_* is the random intercept and *β_j_* is the slope for each patient *j*; and *γ*, *δ*, *η* are dummy coefficients for patients who are of normal weight, overweight, or obese, respectively. For parameter estimation, a full Bayesian model was deployed accounting for the censoring in the response 1 [9,13,14]. The response variable was shifted by one unit to the right prior to applying the log-transformation in order to handle zero values in the response variable and map the minimum, again, to zero.

We next calculated 95% credible (prediction) intervals from the posterior predictive distribution to obtain a range in which one could preemptively calculate the expected RPP volume in a new post-operative RALP patient as a function of their pre-operative BMI. Table 2 depicts the 95% credible intervals for patients in each of the three BMI groups, with POD 3 volumes spanning 0.05 mL (2.5th percentile) to 2.5 L (97.5th percentile) in normally weighted patients, 0.02 mL to 1.27 L in overweight patients, and 0 mL to 154.9 mL in the obese cohort. Exponential reductions in all three groups were seen at PODs 3, 4, 5, 6, and 7, with the model demonstrating narrower intervals at significantly lower RPP volumes in obese patients as opposed to their normal weight and overweight counterparts, the latter of which were similar to one another (no *p*-values possible in full Bayesian models) [14]. For significance testing of the parameter estimates, a range of equivalence of plus or minus 0.1 standard deviations from the general population mean was used, as suggested by Kruschke and Liddell [10]. 

## 4. Discussion

This is the first study to systematically assess RPP after a standardized laparoscopic procedure using CT. In our cohort of 31 post-operative RALP patients, we first quantified the volume of RPP in a local cohort over a seven-day post-operative time course, and later identified pre-operative weight and BMI as the strongest demographic factors affecting RPP volumes.

### 4.1. Description of RPP and Correlation to Demographics

We used a sequence of three low-dose abdominal CT scans within a seven-day post-operative course to show that RPP volumes were, on average, relatively small, from approximately 37 mL on POD 3, 19 mL on POD 5, and 10 mL on POD 7 (Table 1). Not only did we also show a near-exponential resolution of RPP volumes over the post-operative course, but we also noted a large range of RPP volumes (as low as 0 mL and as high as approximately 250 mL), and subsequently correlated this with the significant effects of pre-operative BMI. We then developed a predictive model that incorporated pre-operative weight to predict RPP with 95% credibility.

Though the question of how much RPP to expect post-operatively is a simple one, there is a paucity of literature on the topic despite its significant impact on everyday clinical practice. Only two studies have assessed RPP in the 24 h after surgery: one in the context of laparoscopic cholecystectomy and the other in the context of minor gynecologic surgeries [6,7]. Both studies used either chest or abdominal X-rays to characterize RPP, measured as the height of sub-diaphragmatic air, and both studies demonstrated that an increased RPP volume was closely linked to post-operative pain. Whereas an X-ray can accurately measure the subdiaphragmatic sickle, it tends to mischaracterize the peri-visceral air present intra-abdominally [6,7]. However, a CT carries better sensitivity than an X-ray and is widely regarded as the gold standard for the assessment of post-operative complications [8]. Another imaging modality to assess post-operative residual air is an ultrasound [15], and this is often employed in trauma settings. Again, the CT remains a gold standard and confers superior resolution to ultrasounds.

We also assessed RPP over a longer time course than other published studies, again using a seven-day course as opposed to the first 24 h [6,7]. Prior studies were unable to characterize the time frame of complete RPP resolution, whereas this study could. Moreover, the timing of our post-operative CT scans carries clinical significance, as symptoms from viscous injuries typically do not manifest within the first 24 h but rather within the first three to six days [16,17]. 

An interesting point to note is that the average abdominal volume is approximately nine liters, though it can be as high as twenty in some cases [18]. Most of the patients in our cohort (i.e., those in the third quartile) had RPP volumes below 32 mL, demonstrating that relatively large-volume RPP is rare. The maximum measured volumes for each of the three CT scans were over 200 mL, but such cases were, again, rare. More interesting is the relatively large distribution of RPP volumes as a function of post-op day (3, 5, or 7) and as a function of habitus (normal weight, overweight, and obese), and that obese patients had smaller initial RPP volumes as well as an earlier resolution of RPP. One could argue that the higher proportion of subcutaneous fat in the obese population was itself enough to expel much of the insufflated CO_2_ at the conclusion of surgery. There may also be less space relative to the total abdominal capacity. Lastly, fat has a rich vascular supply and obese men are more likely to carry visceral fat. Thus, obese men with abundant visceral fat may be better able to resorb any RPP volume than their matched normally weighted or overweight counterparts.

### 4.2. Statistical Formula

Based on our data, we also statistically established a predictive model for RRP. Because a statistical formula seems difficult to apply into routine clinical practice, we provided a table (Table 2) for POD 3, 4, 5, 6, and 7 according to the patient’s BMI. If a patient undergoes a CT scan post-operatively, the treating physician can now correlate the RPP with the provided table. This allows one to estimate if RPP at a particular post-operative time point is still in the expected range (i.e., if the volume is below the “upper limit”), or if the volume is too high compared to our reference cohort (i.e., if the volume is higher than the “upper limit”). 

Limitations of this study include its size, scope, and imaging strategy. Though relatively small, the overall size of our cohort is congruent with prior efforts [6,7]. Secondly, we examined RPP in the context of consecutive RALP with no major complications, which represents a relatively standard urological procedure. RPP may vary in surgeries of differing complexity, a more heterogeneous patient population, or in patients with prior abdominal surgical history. No pre-operative CT and no CT later than POD 7 were performed. However, from an ethical perspective, we sought to minimize the radiation exposure per person and the number of individuals exposed to additional radiation post-operatively. Moreover, our present analysis of the RPP is the first study to systematically assess RPP after a standardized laparoscopic procedure using CT, and could serve as a reference for future studies, which should be expanded to include more complex operations in varied populations, centers, and genders. 

## 5. Conclusions

This is the first study to systematically assess RPP after a standardized laparoscopic procedure using CT. One week after RALP, two thirds of patients will exhibit clinically insignificant RPP, even with volumes as high as 250 mL. Larger patients tend to have smaller residuals. Our data provide new knowledge regarding RPP and may help surgeons interpret post-operative CTs in similar patient populations.

## Figures and Tables

**Figure 1 diagnostics-12-00785-f001:**
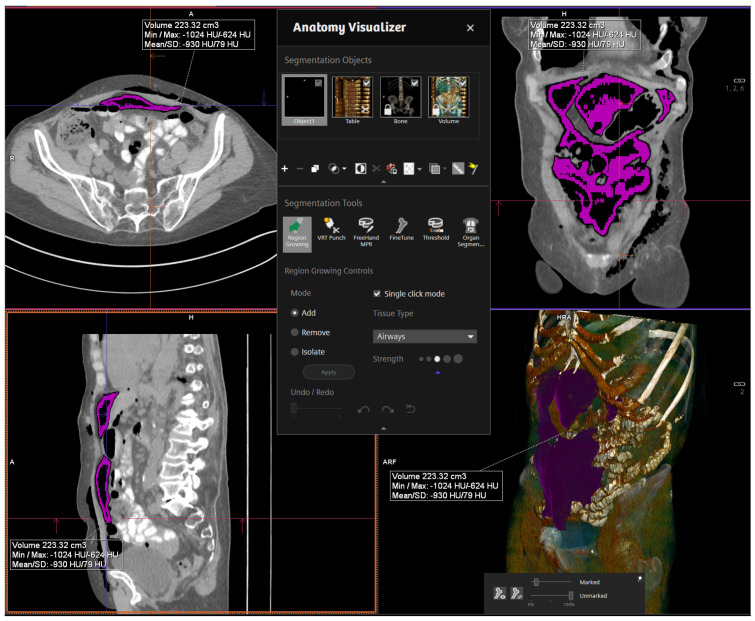
Volume measurement software interface.

**Figure 2 diagnostics-12-00785-f002:**
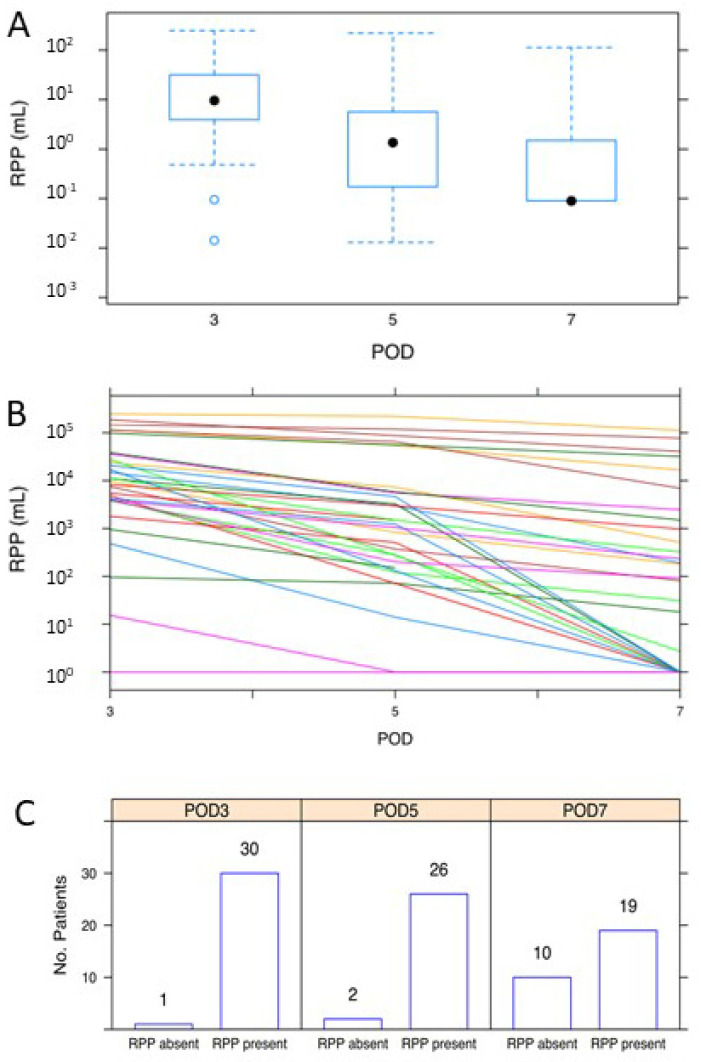
(**A**) RPP at postoperative day 3, 5, and 7; (**B**) RPP for each individual patient; (**C**) number of patients without any detectable RPP.

**Figure 3 diagnostics-12-00785-f003:**
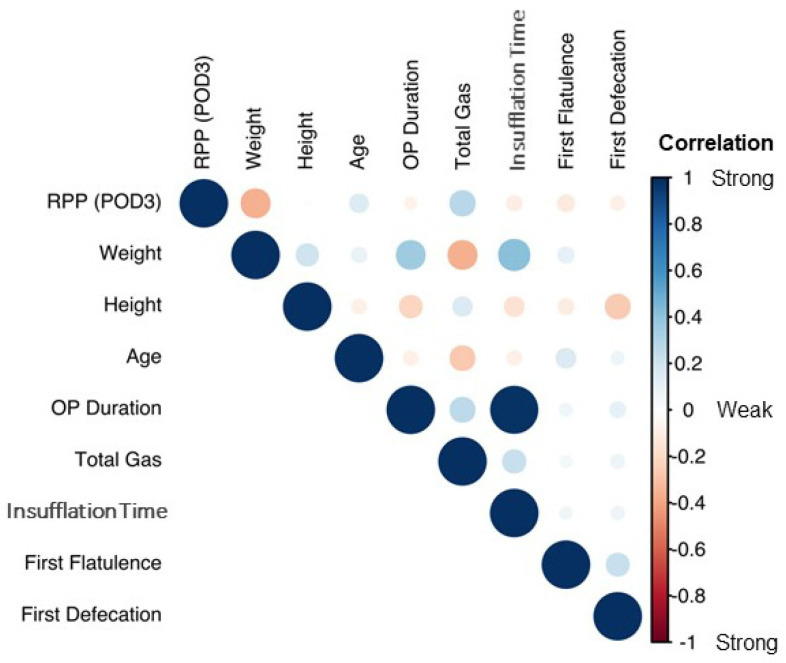
Correlation plot with demographic and operative data. Light color/circular shape: weak correlation; strong color/elliptical shape: strong correlation.

**Figure 4 diagnostics-12-00785-f004:**
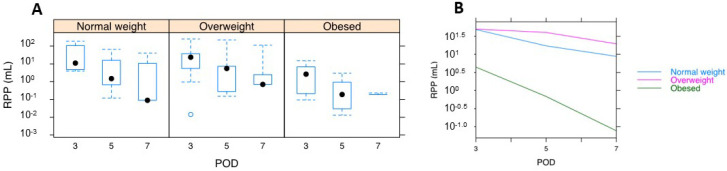
Patient RPP volumes are plotted as a function of stratified BMIs separately ((**A**): normal, overweight, obese) and combined (**B**).

**Table 1 diagnostics-12-00785-t001:** Summary of the patient demographics and operative parameters.

	*n*	Mean	SD	Median	IQR
Age	31	66.2	6.7	67	62–70.5
Weight (kg)	31	83.3	12.4	83	75.5–90.5
Height (cm)	31	177.2	6.1	178	173–180.5
BMI	31	26.6	4.1	26.0	24.1–29.2
OP Duration (min)	31	302.4	43.4	306	270–328
Total Gas (mL)	29	971,448	426,203	941,000	775,000–1,242,000
Insufflation Time (min)	30	278.9	42.3	276.5	255–304.2
RPP POD 3 (mL)	31	37.6	61.8	9.6	3.9–31.8
RPP POD 5 (mL)	28	19.8	48.1	1.4	0.2–5.6
RPP POD 7 (mL)	29	10.1	25.8	0.1	0.0–1.5

RPP: residual pneumoperitoneum; POD: postoperative day; SD: standard deviation; IQR: interquartile range.

**Table 2 diagnostics-12-00785-t002:** Calculated RPP at post-operative day 3, 4, 5, 6, and 7 showing 95% credible intervals for patients in each of the three BMI groups.

Postoperative Day	3	4	5	6	7
Normal Weight Patient (RPP in mL)					
Upper limit (97.5%)	2500.95	1523.20	856.85	573.02	411.99
Lower limit (2.5%)	0.05	0	0	0	0
Overweight Patient (RPP in mL)					
Upper limit (97.5%)	1271.42	748.61	472.70	315.49	219.31
Lower limit (2.5%)	0.02	0	0	0	0
Obese Patient (RPP in mL)					
Upper limit (97.5%)	154.94	87.48	45.96	29.52	24.46
Lower limit (2.5%)	0	0	0	0	0

RPP: residual pneumoperitoneum.

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
