# Peer review of "Quantifying and Statistically Modeling Residual Pneumoperitoneum after Robotic-Assisted Laparoscopic Prostatectomy: A Prospective, Single-Center, Observational Study"

_diagnostics, 2022, doi:10.3390/diagnostics12040785_

Round 1

Reviewer 1 Report

All references should be indexed in text before punctuation. 

More references should be added, in the introductory and discussion parts. 

The authors should mention another imagistic methods used to detect pneumoperitoneum, like abdominal ultrasound. Cite: Socea B, et al. PNEUMOPERITONEUM DIAGNOSED USING ULTRASONOGRAPHY - A NARRATIVE REVIEW OF THE LITERATURE. RST, 2019, 1(17): 219-223. 

Reviewer 2 Report

Dear Authors,

I reviewed with interest the paper entitled “Quantifying and statistically modeling residual pneumoperitoneum after robotic-assisted laparoscopic prostatectomy: a prospective, single-center, observational study”, aiming to prospectively quantify the residual pneumoperitoneum (RPP) after robotic-assisted laparoscopic prostatectomy (RALP) by analyzing 31 cases of RALP through TC scans on POD 3, 5 and 7.

I found the present study interesting, well written and fluent to read - just minor English editing is required. It concerns with an actual topic, due to the wide use of robot-assisted laparoscopic approach in several surgical areas.

- The title is descriptive of what authors have explored in their work.

- The background and scientific rationale for carrying out the study are well presented. Yet, especially due to the wide use of robot-assisted laparoscopic approach in several surgical areas and especially in urologic oncology - as mentioned in lines 1-2 -, I would suggest adding some references to support this statement for each area (e.g., PMID: 34038044; doi: 10.1007/s00423-022-02465-0; doi: 10.1007/s11701-019-00953-y; DOI: 10.5173/ceju.2021.0017.R3).

- Study design is clearly stated.

- As reported in paragraph 2.3 Authors evaluated whether manual abdominal decompression at the end of the case was performed or not, yet I do not see this variable in Figure 3. Did you considered this variable - that could have a high impact on outcomes - in your analysis? Please clarify

- Paragraph 2.5 should be improved. I would suggest moving Ethics in Study design paragraph and expand Statistics one. Indeed, all the analyses performed - which are now missing - should be added and reported in this paragraph.

- Figures are clear. Tables as well as Results section could be improved, specifically regarding Table1 and paragraph 3.1 (they are a bit repetitive: please do not put all the data in the text and modify/move columns in Table 1 to be more clear - e.g., use mean (SD), median (range or IQR)).

- Overall, Discussion is adequately implemented with the relevant literature, and interpretations and conclusions are well stated and justified by results. I would suggest just mitigate the second sentence of the first paragraph.

I have not further comments. 
